# Secular Trend of Self-Concept in the Context of Childhood Obesity—Data from the CHILT III Programme, Cologne

**DOI:** 10.3390/children10010127

**Published:** 2023-01-07

**Authors:** Lisa Grünberg, Nina Eisenburger, Nina Ferrari, David Friesen, Fabiola Haas, Marlen Klaudius, Lisa Schmidt, Christine Joisten

**Affiliations:** 1Department for Physical Activity in Public Health, Institute of Movement and Neurosciences, German Sport University Cologne, Am Sportpark Müngersdorf 6, 50933 Cologne, Germany; 2Cologne Center for Prevention in Childhood and Youth, Heart Center Cologne, University Hospital of Cologne, Kerpener Str. 62, 50937 Cologne, Germany

**Keywords:** childhood obesity, psychosocial distress, secular trend, self-concept, physical appearance, physical fitness

## Abstract

Juvenile obesity is associated with a lower self-concept. Given the continued prevalence of obesity, we examined the secular trend of body mass index standard deviation score (BMI-SDS) and self-concept in participants of a German weight management programme (*n* = 242, 53.3% female, mean age 12.5 ± 2.1 years, mean BMI-SDS 2.45 ± 0.46) over a period of 15 years. Anthropometric data, physical fitness (watt/kg), and demographic data were assessed. The years 2005 to 2020 were grouped into six sections containing a mean of *n* = 40 participants. The questionnaire for the assessment of self and competence in children (FSK-K) was used to assess the following domains: “scholastic competence”, “social competence”, “physical appearance”, “global self-worth”, and “behavioural conduct”. No significant between-group differences in self-assessment across self-concept domains were found. In all time periods, heavier children assigned the lowest rating to physical appearance. Social competence increased with higher physical fitness. Even though no negative trend in the self-concept of children with obesity was found in this cohort, the findings confirmed an association between juvenile overweight/obesity and lower physical self-concept, and between a better social competence and increasing physical fitness. Accompanying psychosocial care, therefore, rightly remains an important pillar of obesity therapy.

## 1. Introduction

In 2016, 124 million children (50 million girls and 74 million boys) were obese worldwide, while another 123 million children were overweight [1]. In Germany, 15.4% of children were overweight and 5.9% of them were obese between 2014 and 2017, according to KiGGS Wave 2 (“Children and Adolescents Health Survey”; authors translation) of the Robert Koch Institute [2]. The recent coronavirus pandemic (COVID-19) has exacerbated this trend, especially in families with lower socioeconomic status and particularly in children and adolescents who were already overweight pre-pandemic [3]. Another German study confirmed these findings, indicating that during the pandemic there was excess weight gain in every sixth child [4].

In addition to physical co-morbidities, such as arterial hypertension, elevated lipids [5,6,7], and the presence of insulin resistance—even manifested as type-2 diabetes [8]—affected children and adolescents can be particularly burdened by psychosocial issues. Thus, 40% suffer from anxiety disorders, 15% have somatization disorders, and 17% suffer from eating disorders [9]. Additional issues include low self-esteem [10], discrimination and stigmatization [11], social exclusion and deficits in social skills [12,13], bullying [14], difficulties in school matters (for example, negative attitudes towards children with higher bodyweight were observed among teachers) [15], other psychological disorders, and body dissatisfaction, as well as lower health-related quality of life [13]. Topçu et al. [16] applied the Piers–Harris children’s self-concept scale and found that children who were obese experienced more psychiatric disorders than their peers without obesity. In sum, weight-related stressors lead to increased social withdrawal, further weight gain [17], and depressed mood [18], and manifest as depression in 10.4% of cases [19].

Given the increasing rates of obesity, it is of interest whether this is accompanied by a parallel increase in psychosocial distress. However, data regarding the secular trend of mental comorbidities accompanying juvenile obesity are sparse.

Family-based multicomponent weight management programmes, with behavioural, dietary, and physical activity components, are the best practice for achieving successful weight loss and alleviating the associated psychosocial burden in affected children and adolescents [20]. However, if the prevalence of psychosocial problems has increased in parallel with growing obesity rates in recent years, programmes may need to be adapted to address this problem.

The primary aim of this analysis was therefore to investigate whether the body mass index standard deviation score (BMI-SDS) and self-concepts of children and adolescents participating in a weight management programme have changed over the past 15 years. As a secondary objective, we examined predictors of self-concept domains across different time spans to uncover any trends.

## 2. Materials and Methods

### 2.1. Population Sample/Study Design

Data from the children’s health interventional trial (CHILT) III, an outpatient, multimodal, family-based weight management programme for children and adolescents aged eight to sixteen years [21] (*n* = 535), were examined. The CHILT Project is registered in the German Clinical Trials Register under ID DRKS00026785. To be included in the study, complete records on sex, age, height, and weight, as well as the FSK-K questionnaire (questionnaire for the assessment of self and competence in children) of the participants had to be available (*n* = 242; Figure 1).

The years between 2005 and 2020 were divided into six time periods of two to three years each. Each time span or period contained a similar number of study participants, with a mean of *n* = 40, a maximum of *n* = 45 (2015–2017) and a minimum of *n* = 34 (2009–2011) participants per group. (Table 1) The first period of available data from 2003 to 2004 was excluded from the evaluation due to insufficient data.

Of the *n* = 242 participants, 46.7% were male (*n* = 113) and 53.3% were female (*n* = 129). Participants were on average 12.5 ± 2.1 years old, had a mean height of 1.59 ± 0.11 m, weighed an average of 76.4 ± 19.6 kg, and had a mean BMI of 30.0 ± 4.8 kg/m^2^. The mean BMI-SDS was 2.45 ± 0.46. Overall, 55.8% were younger than 13 years of age (pre-adolescent), while 44.2% were 13 years of age or older (adolescent) [22]. Sex differences can be found in Table 1. Descriptive statistics of all time clusters separated by sex can be found in the Appendix A.

### 2.2. Anthropometric Data

The height and weight (including light sportswear) of the study participants were measured barefoot. Standard calibrated scales were used (weight: Seca^®^ scale, type 225; height Seca^®^ scale, type 761). Following Kromeyer-Hauschild et al., the children were classified according to German percentiles, i.e., those in the 90% percentile and above were classified as overweight, while those in the 97% percentile and above were classified as obese [23]. BMI standard deviation scores (BMI-SDS) were expressed age- and sex-specifically according to the LMS method for non-normally distributed sizes. M, L, and S were the individual parameters for the child’s age and sex.

Adolescent status was distinguished by age: below 13 years (pre-adolescent) or 13 years and above (adolescent) [24].

### 2.3. Demographics and Media Consumption

Demographic data, socioeconomic status, and information on family lifestyle were requested using a standardized parental questionnaire [25,26]. Migration background was determined by the language spoken at home (yes or no to speaking German) [27].

Media consumption (hours per day) was assessed by asking parents to estimate their child’s total daily engagement with television, video games, internet use, and mobile phone, as well as time spent listening to music.

### 2.4. Self-Concept

The assessment of self-concept was completed using the “Questionnaire for the Assessment of Self and Competence for Children (FSK-K)” by Wünsche and Schneewind, a German version of Harter’s Self-Perception Profile for Children (SPPC) [28,29]. The 30 items were divided into five domains of six questions each: “scholastic competence”, “social competence”, “physical appearance”, “global self-worth”, and “behavioural conduct”. The items were presented as positively or negatively worded statements. Each participant had to choose between “true for me” or “somewhat true for me” for each statement. The answers were coded from 1 to 4, with 4 always representing the most positive response possible. For example, the statement “I have many friends” could be answered with “true for me” (coded 4) or “somewhat true for me” (coded 3). This statement’s counterpart, “I do not have many friends”, could be answered with “somewhat true for me” (coded 2) or “true for me” (coded 1). The data were then recoded, with the highest domain-specific competence defined as 100 on the mean scale. Thus, the participants’ self-concept ratings could be compared across domains using their mean values. The internal consistencies (Cronbach’s alpha) of the self-concept domains were 0.79 for “scholastic competence” (*n* = 234), 0.82 for “social competence” (*n* = 225), 0.76 for “physical appearance” (*n* = 218), 0.77 for “behavioural conduct” (*n* = 231), and 0.71 for “global self-worth” (*n* = 216).

### 2.5. Physical Fitness

Physical fitness was calculated as the relative wattage, i.e., watts per kilogram of body weight. The physical fitness measure “Peak Mechanical Power (PMP [watt])” was determined via bicycle ergometry (Ergoline^®^ ergometrics 900). For this purpose, the participants pedalled to exhaustion, with testing starting at 25 watts and increasing by 25 watts every two minutes (WHO (World Health Organization) scheme) [30]. Discontinuation criteria included, but were not limited to, clinical symptoms (such as dizziness or a severe headache), abnormalities during medical monitoring (such as non-physiological abnormalities in an ECG or blood pressure), and technical problems [31].

### 2.6. Statistical Analysis

A repeated cross-sectional design was used. This study derives from a post-hoc analysis of the data collected in a weight management programme. Therefore, the sample size was not established since the aims of the present study, and we performed a post-hoc power calculation. The post-hoc power analysis was performed with G*Power version 3.1.9.6 (Heinrich-Heine-Universität Düsseldorf, Germany). At least 177 participants were required for this study to perform an analysis of covariance (ANCOVA) comparing six groups with f^2^ = 0.35, a power of 0.95 at an alpha level of 0.05, five degrees of freedom and four covariates [32].

All analyses were performed using the IBM programme SPSS Statistics version 27.0 (IBM Corp., Armonk, NY, USA). Categorical variables were expressed as percentages and frequencies, while continuous variables were expressed in terms of means, standard deviations (SD), maximums (max), and minimums (min). Unpaired *t*-tests and chi-squared tests were used to detect possible differences between sexes.

Analysis of covariance (ANCOVA) was used to analyse differences between the six time periods between 2005 and 2020 in the five domains of self-concept as well as BMI-SDS as dependent variables.

In addition, moderation analyses were run to determine whether the interaction between BMI-SDS and time significantly predicted the subdomains of self-concept. For this purpose, multiple linear regression analyses were performed including the mean centred variables of BMI-SDS and time and their interaction effect while controlling for age, adolescent status, and sex.

Lastly, a backward stepwise linear regression was performed to examine factors that were consistently associated with the domains of self-concept across time spans. Sex (m/f), age, BMI-SDS, and adolescent status (<13/≥13 years) as well as migration background (yes/no), media consumption (h/day), and physical fitness (watts/kg) were included as predictors in the baseline models. Only significant variables were included in the final model, as insignificant variables were removed step by step. A *p*-value of <0.05 was considered significant.

## 3. Results

### 3.1. Secular Trends in Anthropometric Data

The analysis showed no significant change in height, weight, BMI, and BMI-SDS over time among children who were overweight and participating in a weight management programme (Table 2, Figure 2). The same analysis separated by sex is shown in the Appendix A.

### 3.2. Secular Trends in Self-Concept

With regard to the secular trend in self-concept, we could not find any significant differences (Table 3, Figure 3).

Analysing differences in the domains of self-concept over time separated by sex is shown in the Appendix A. Additional calculations in terms of self-concept domains considering migration background, media consumption, and physical fitness are shown in the Appendix A.

Examining the interaction effect of time and BMI-SDS on the subdomains of self-concept, we did not find significance (all results had *p* > 0.5), except in the physical appearance model. The results showed that time significantly moderated the effect between BMI-SDS and perceived physical appearance; adj. R^2^ = 0.16, F (6, 236) = 8.4, *p* < 0.001.

### 3.3. Consistent Predictors of Self-Concept across Time Spans

In four of six time clusters of the physical appearance domain, a lower BMI-SDS was associated with a higher “physical appearance” rating in the final models (ß = −0.420, *p* = 0.005 in 2007–2008; ß = −0.605, *p* = 0.007 in 2009–2011; ß = −0.651, *p* = 0.001 in 2012–2014, and ß = −0.391, *p* = 0.014 in 2015–2017; Table 4).

Regarding the scholastic competence domain across time periods, the final models built with backward stepwise linear regression did not identify any consistent factors (Table 5). Only adolescent status (2009–2011 (ß = 0.801, *p* = 0.009) and 2012–2014 (ß = −0.456, *p* = 0.009)) and physical fitness (2005–2006 (ß = 0.474, *p* = 0.004) and 2007–2008 (ß = 0.512, *p* = 0.027)) had significant influences on this domain in more than one time span.

In the social competence domain, higher physical fitness was associated with better social competence in almost all final models except those of 2005–2006 and 2009–2011 (ß = 0.340, *p* = 0.042 in 2007–2008; ß = 0.347, *p* = 0.009 in 2012–2014; ß = 0.536, *p* = 0.001 in 2015–2017 and ß = −0.481, *p* = 0.010 in 2018–2020) (Table 6).

After accounting for all other variables, BMI-SDS remained a predictor in the final models of the global self-worth domain in the years of 2007–2008 (ß = −0.247, *p* = 0.140), 2012–2014 (ß = −0.727, *p* < 0.001) and 2015–2017 (ß = 0.433, *p* = 0.006) (Table 7).

In the behavioural conduct domain, migration background remained in the final models in 2012–2014 (ß = −0.396, *p* = 0.029), 2015–2017 (ß = −0.402, *p* = 0.011) and 2018–2020 (ß = −0.252, *p* = 0.117) (Table 8).

## 4. Discussion

The prevalence of overweight and obesity, or high weights, among children and adolescents is increasing worldwide [1,33]. This has significant consequences for physical and mental health [34,35,36], which may be relevant within weight management programmes. Therefore, the current study collated and analysed data on BMI-SDS as well as on five subdomains of the self-concept in children and adolescents who participated in a German weight-management programme between 2005 and 2020. Contrary to global trends, this sample showed that BMI-SDS remained stable during this period. Although there were shifts in favour of younger and male participants, these trends were not significant. In addition, the results did not suggest either an improvement or a decline in the self-concept of overweight youths over time. The domain “physical well-being” was consistently rated with the lowest values. Furthermore, BMI-SDS was found to negatively correlate with almost all domains of self-concept, especially physical self-concept, at all different points in time. However, this negative correlation between BMI-SDS and physical appearance became weaker over time. The possible influence of additional factors such as migration background, physical fitness, media consumption, adolescent status, age, and gender on the different domains of the self-concept varied over the years. In the domains “physical appearance” and “global self-worth”, a higher BMI-SDS was most often associated with a lower score. In the domains “scholastic competence” and “behavioural conduct” no clear factor was found, whereas in the domain of “social competence,” higher physical fitness was associated with a better assessment in almost all final models.

In earlier studies, Ottova et al. [36] described the effects of being overweight on the health-related quality of life of children and adolescents in a European cohort with 17,159 participants between the ages of 8 and 18. Consistent with the current findings, they found that the most impaired areas of health-related quality of life were physical well-being and self-perception. Older children and adolescents tended to have a negative self-image and a poorer assessment of their own bodies [11,12,37]. In general, girls appear to be more dissatisfied with their bodies and thus more susceptible to possible psychological disorders, such as eating disorders [11,37,38]. However, Molina-García et al. [39] found higher BMI to be associated with body dissatisfaction in both sexes.

In contrast, physical activity (or fitness as a proxy) has already been associated with better physical and mental health. For example, the self-concept of normal-weight children and adolescents was positively associated with physical activity in Babic et al. [40] and with cardiovascular fitness in Vedul-Kjelsås et al. [41]. Prior research has also found a significant association between physical activity and physical self-concept in normal-weight children and adolescents, with age and gender being key moderators of this association [40]. This association has also been observed among children and adolescents with overweight and obesity [42,43]. In their study, Perez-Sousa et al. [42] demonstrated an association between the Z-scores of physical fitness and several HRQoL dimensions (i.e., the physical, emotional, social, and psychosocial dimensions) in overweight children and adolescents. It was only for the dimension of school competence that no correlation was found. In our data, however, there was a significant correlation between physical fitness and school and social competence. It is possible that better fitness ensures a higher ability to connect, and prevents social withdrawal in the school environment. In a systematic review, Silva et al. (2018) showed that increasing fitness was one motivating factor for weight loss in adolescents with overweight or obesity in at least 2 of 6 integrated studies. However, the main motive out of 17 mentioned factors was the desire for health improvements followed by an increase of self-esteem and avoidance of bullying [44]. In an earlier analysis, we already proved the connection between physical fitness and quality of life or self-concept [43]. Improving physical fitness may be more promising for positive mental health outcomes in weight management programmes than weight loss or participation in physical activities alone.

## 5. Strengths and Limitations

One of the strengths of this study is the continuous and standardized recording of parameters over almost two decades. For example, height, weight, BMI-SDS, physical fitness, and migration background of each participant were measured by qualified personnel. The FSK-K questionnaire is also considered a standardized instrument for assessing self-concept in children and has been used several times. However, these measurements might be considered as only snapshots in time, rather than comprehensive. In addition, we cannot exclude the possibility of distorted data resulting from self-reporting. Furthermore, our collective consisted of small monocentric groups in the different year clusters. Finally, potential confounding factors such as family structure or diet could not be integrated into the analysis. Thus, the generalisability of the current results is limited.

## 6. Conclusions

Considering the limitations, the available data do not suggest any trends within the individual psychosocial self-concept domains and the BMI-SDS for children and adolescents with obesity over the past decades. However, in most time clusters, the domains “physical appearance” and “social competence” correlated negatively with BMI-SDS and positively with physical fitness. On the one hand, these results therefore underline the well-known importance of psychosocial support as a therapy pillar in weight management programmes. On the other hand, more attention should also be paid to the promotion of physical fitness, as it could contribute not only to the reduction of BMI-SDS but also to an improvement of the participants’ self-concepts. To corroborate the current findings, future multicentred research should undertake analysis on larger groups of participants.

## Figures and Tables

**Figure 1 children-10-00127-f001:**
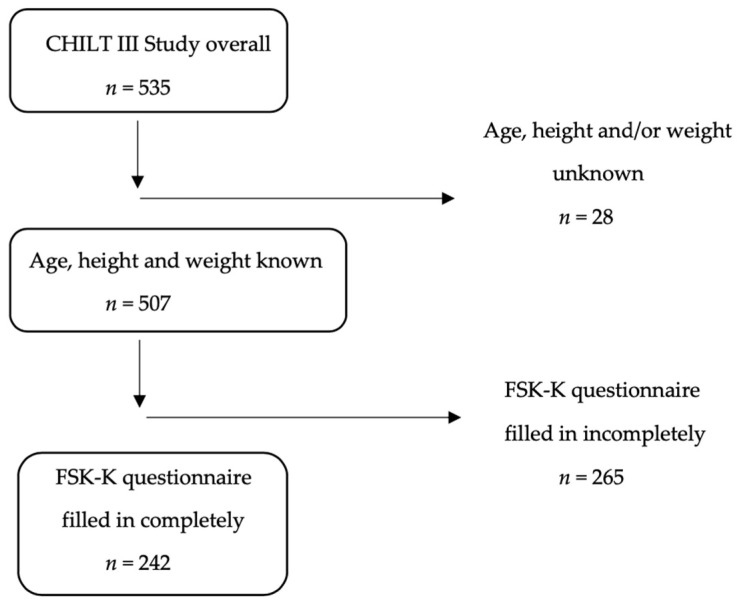
Inclusion criteria for the study population.

**Figure 2 children-10-00127-f002:**
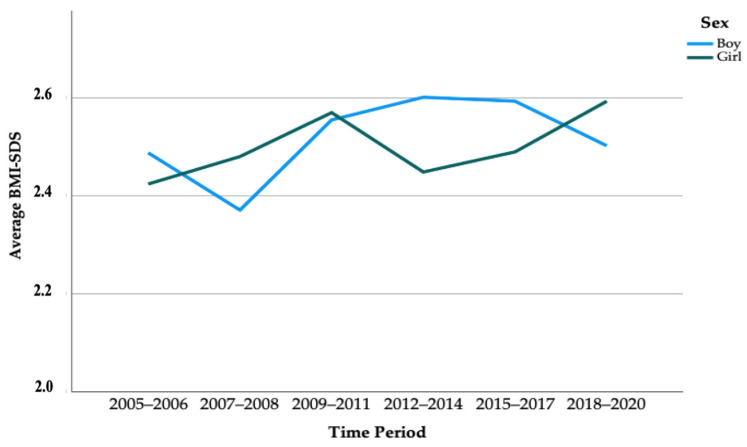
Secular trend of average BMI-SDS between 2005 and 2020 separated by time periods and sex. *p* = 0.631 according to an analysis of covariance (ANCOVA).

**Figure 3 children-10-00127-f003:**
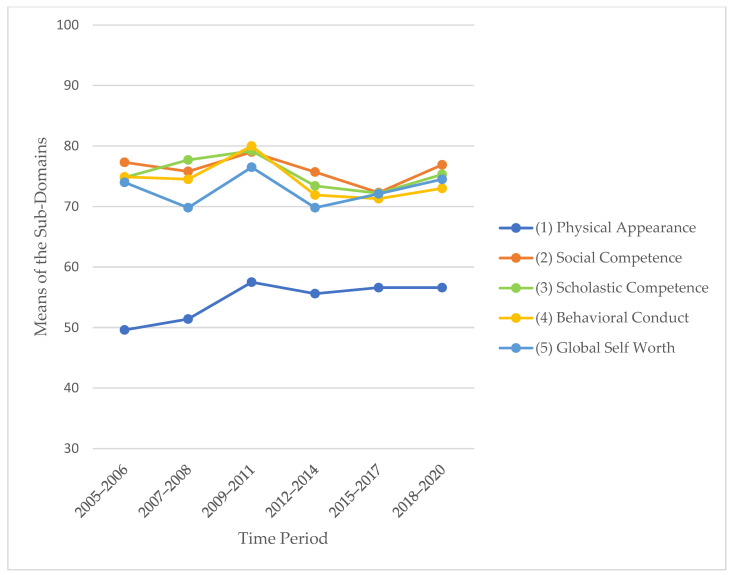
Means of the five sub-domains of self-concept (recoded data scaled 0–100) in the different time periods.

**Table 1 children-10-00127-t001:** Population characteristics.

	Total (*n* = 242)	Boys (*n* = 113)	Girls (*n* = 129)	*p*-Value
(1) Sex *n* (%)		113 (46.7%)	129 (53.3%)	
(2) Age (in years)	12.5 ± 2.1 (242)	12.7 ± 2.1 (113)	12.3 ± 2.2 (129)	0.110 ^†^
(3) Height (in m)	1.59 ± 0.11 (242)	1.62 ± 0.12 (113)	1.56 ± 0.10 (129)	<0.001 ^†^
(4) Weight (in kg)	76.4 ± 19.6 (242)	81.3 ± 22.3 (113)	72.1 ± 15.8 (129)	<0.001 ^†^
(5) BMI (in kg/m^2^)	30.0 ± 4.8 (242)	30.7 ± 5.4 (113)	29.4 ± 4.2 (129)	0.038 ^†^
(6) BMI-SDS	2.45 ± 0.46 (242)	2.45 ± 0.47 (113)	2.45 ± 0.45 (129)	0.974 ^†^
(7) Adolescent status				
<13 years old	135 (55.8%)	57 (50.44%)	78 (60.47%)	
≥13 years old	107 (44.2%)	56 (49.56%)	51 (39.53%)	0.117 ^‡^
	Overall (*n* = 205)	Boys (*n* = 97)	Girls (*n* = 108)	
(8) Migration background (yes)	23 (11.2%)	10 (10.3%)	13 (12.0%)	0.696 ^‡^
(9) Media consumption (h/day)	3.23 ± 3.27 (205)	3.45 ± 3.82 (97)	3.02 ± 2.69 (108)	0.349 ^†^

The data are presented as the mean ± SD; BMI = body mass index; BMI SDS = body mass index standard deviation score; *p*-values calculated with the ^†^ unpaired *t*-test; ^‡^ Chi square test.

**Table 2 children-10-00127-t002:** Descriptive statistics separated by time period.

Overall (*n* = 242)	2005–2006 (*n* = 40)	2007–2008 (*n* = 38)	2009–2011 (*n* = 34)	2012–2014 (*n* = 42)	2015–2017 (*n* = 45)	2018–2020 (*n* = 43)	*p*-Value
(1) Sex *n* (%)							
male	15 (37.5%)	19 (50.0%)	14 (41.2%)	22 (52.4%)	20 (44.4%)	23 (53.5%)	0.642 ^‡^
female	25 (62.5%)	19 (50.0%)	20 (58.8%)	20 (47.6%)	25 (55.6%)	20 (46.5%)	
(2) Age (in years)	12.3 ± 2.1	12.9 ± 2.0	12.6 ± 1.6	12.4 ± 2.5	12.4 ± 2.2	12.4 ± 1.9	0.835 ^†^
(3) Height (in m)	1.58 ± 0.11	1.62 ± 0.13	1.57 ± 0.09	1.59 ± 0.12	1.56 ± 0.12	1.60 ± 0.11	0.233 ^†^
(4) Weight (in kg)	74.4 ± 19.3	78.2 ± 20.5	73.7 ± 15.0	77.8 ± 18.6	74.4 ± 22.0	79.7 ± 21.2	0.670 ^†^
(5) BMI (in kg/m^2^)	29.5 ± 4.9	29.4 ± 4.3	29.7 ± 4.1	30.4 ± 4.5	30.1 ± 6.1	30.5 ± 4.7	0.867 ^†^
(6) BMI-SDS	2.41 ± 0.51	2.37 ± 0.40	2.41 ± 0.46	2.52 ± 0.49	2.46 ± 0.46	2.51 ± 0.44	0.631 ^†^
(7) Adolescent status							
<13 years old	24 (60.0%)	20 (52.6%)	17 (50.0%)	21 (50.0%)	26 (57.8%)	27 (62.8%)	0.791 ^‡^
≥13 years old	16 (40.0%)	18 (47.4%)	17 (50.0%)	21 (50.0%)	19 (42.2%)	16 (37.2%)	
Overall (*n* = 205)	2005–2006 (*n* = 31)	2007–2008 (*n* = 36)	2009–2011 (*n* = 29)	2012–2014 *(n* = 30)	2015–2017 (*n* = 39)	2018–2020 (*n* = 40)	
(8) Migration background (yes)	2 (6.4%)	3 (8.3%)	1 (3.4%)	6 (20%)	3 (7.7%)	8 (20%)	0.616 ^‡^
(9) Media consumption (in h/day)	1.90 ± 1.11	1.67 ± 0.96	1.35 ± 0.95	1.44 ± 0.89	5.25 ± 3.65	6.38 ± 4.11	<0.001 ^†^

The data are presented as the mean ± SD; BMI = body mass index; BMI SDS = body mass index standard deviation score; *p*-values calculated with the ^†^ analysis of variance (ANOVA); ^‡^ chi square test.

**Table 3 children-10-00127-t003:** Results analysing differences in the domains of self-concept over time.

		Periods of Analysis						
	Overall (*n* = 242)	2005–2006 (*n* = 40)	2007–2008 (*n* = 38)	2009–2011 (*n* = 34)	2012–2014 (*n* = 42)	2015–2017 (*n* = 45)	2018–2020 (*n* = 43)	*p*-Value
(1) Physical appearance	54.6 ± 15.7	49.6 ± 14.5	51.4 ± 15.9	57.5 ± 12.3	55.6 ± 15.7	56.6 ± 17.1	56.6 ± 16.9	0.063 ^†^
(2) Social competence	76.0 ± 18.5	77.3 ± 19.2	75.8 ± 21.7	79.0 ± 18.5	75.7 ± 15.4	72.3 ± 19.2	76.9 ± 17.4	0.713 ^†^
(3) Scholastic competence	75.2 ± 16.6	74.8 ± 19.5	77.7 ± 14.8	79.2 ± 14.5	73.4 ± 15.8	72.2 ± 15.9	75.3 ± 18.2	0.466 ^†^
(4) Behavioural conduct	74.0 ± 16.3	74.9 ± 17.1	74.5 ± 16.8	80.0 ± 15.5	71.9 ± 17.6	71.3 ± 13.8	73.0 ± 16.2	0.269 ^†^
(5) Global self-worth	72.7 ± 16.0	74.0 ± 16.8	69.8 ± 16.9	76.5 ± 12.8	69.8 ± 14.0	72.1 ± 18.1	74.5 ± 16.2	0.377 ^†^

The data are presented as the mean ± SD; *p*-values are calculated with an ^†^ analysis of covariance (ANCOVA) adjusting for sex, age, adolescent status and BMI-SDS.

**Table 4 children-10-00127-t004:** Linear regression analysis (backward) of the physical appearance domain.

Domain	Periods of Analysis of Final Model		Unstandardized Coefficients		Standardized Coefficients	T	*p*-Value	95.0% Confidence Interval for B		Adjusted R^2^
			B	std. error	ß			lower limit	upper limit	
Physical appearance	2005–2006	media consumption	−6.510	2.204	−0.481	−2.954	0.006 *	−11.02	−2.00	0.205
	2007–2008	age	−3.728	1.186	−0.466	−3.143	0.004 *	−6.14	−1.32	0.423
		BMI-SDS	−15.173	4.992	−0.420	−3.040	0.005 *	−25.33	−5.02	
		media consumption	6.150	2.408	0.357	2.553	0.015 *	1.25	11.05	
	2009–2011	BMI-SDS	−17.225	5.840	−0.605	−2.950	0.007 *	−29.23	−5.22	0.226
		physical fitness	−27.541	10.140	−0.557	−2.716	0.012 *	−48.39	−6.70	
	2012–2014	age	−6.791	1.972	−0.988	−3.443	0.002 *	−10.85	−2.74	0.343
		adolescent status	22.076	9.294	0.648	2.375	0.025 *	2.97	41.18	
		BMI-SDS	−23.201	6.178	−0.651	−3.756	0.001 *	−35.90	−10.50	
	2015–2017	BMI-SDS	−13.787	5.334	−0.391	−2.585	0.014 *	−24.59	−2.98	0.130
	2018–2020	physical fitness	−7.611	3.937	−0.299	−1.933	0.061	−15.58	0.36	0.066

Final models of the backward linear regression analysing sub-domains of self-concept as dependent variable and the independent factors sex, age, BMI-SDS, adolescent status (<13 or ≥13 years old), migration background, media consumption in h/day, and physical fitness in watts/kg bodyweight in different time periods (*n* = 205). * significant results (*p* < 0.05).

**Table 5 children-10-00127-t005:** Linear regression analysis (backward) of the scholastic competence domain.

Domain	Periods of Analysis of Final Model		Unstandardized Coefficients		Standardized Coefficients	T	*p*-Value	95.0% Confidence Interval for B		Adjusted R^2^
			B	std. error	ß			lower limit	upper limit	
Scholastic competence	2005–2006	sex	−10.928	5.147	−0.315	−2.123	0.043 *	−21.49	−0.37	0.350
		adolescent status	−10.386	5.283	−0.296	−1.966	0.060	−21.23	0.45	
		physical fitness	23.025	7.274	0.474	3.165	0.004 *	8.10	37.95	
	2007–2008	BMI-SDS	13.381	7.482	0.398	1.788	0.083	−1.83	28.59	0.086
		physical fitness	19.616	8.512	0.512	2.305	0.027 *	2.32	36.91	
	2009–2011	age	−6.088	2.614	−0.654	−2.329	0.028 *	−11.47	−0.70	0.294
		adolescent status	22.953	8.128	0.801	2.824	0.009 *	6.21	39.69	
		migration background	−26.634	12.689	−0.341	−2.099	0.046 *	−52.77	−0.50	
	2012–2014	adolescent status	−14.536	5.189	−0.456	−2.801	0.009 *	−25.18	−3.89	0.236
		migration background	12.496	6.428	0.316	1.944	0.062	−0.69	25.69	
	2015–2017	BMI-SDS	−6.879	5.124	−0.216	−1.343	0.188	−17.26	3.50	0.021
	2018–2020	age	1.317	1.479	0.143	0.891	0.379	−1.68	4.31	−0.005

Final models of the backward linear regression analysing sub-domains self-concept as dependent variable and the independent factors sex, age, BMI-SDS, adolescent status (<13 or ≥13 years old), migration background, media consumption in h/day, and physical fitness in watts/kg bodyweight in different time periods (*n* = 205). * significant results (*p* < 0.05).

**Table 6 children-10-00127-t006:** Linear regression analysis (backward) of the social competence domain.

Domain	Periods of Analysis of Final Model		Unstandardized Coefficients		Standardized Coefficients	T	*p*-Value	95.0% Confidence Interval for B		Adjusted R^2^
			B	std. error	ß			lower limit	upper limit	
Social competence	2005–2006	media consumption	−4.315	2.863	−0.269	−1.507	0.143	−10.17	1.54	0.041
	2007–2008	physical fitness	19.640	9.314	0.340	2.109	0.042 *	0.71	38.57	0.090
	2009–2011	BMI-SDS	13.887	7.460	0.337	1.862	0.074	−1.42	29.19	0.081
	2012–2014	age	−3.797	1.572	−0.617	−2.415	0.023 *	−7.04	−0.56	0.329
		adolescent status	17.552	7.731	0.575	2.270	0.032 *	1.63	33.47	
		physical fitness	12.211	5.378	0.347	2.271	0.032 *	1.14	23.29	
		media consumption	8.313	2.684	0.480	3.097	0.005 *	2.79	13.84	
	2015–2017	adolescent status	−11.492	6.633	−0.284	−1.733	0.092	−24.96	1.97	0.271
		physical fitness	31.041	8.147	0.536	3.810	0.001 *	14.50	47.58	
		media consumption	1.793	0.917	0.325	1.955	0.059	−0.07	3.65	
	2018–2020	BMI-SDS	−12.029	6.762	−0.317	−1.779	0.083	−25.73	1.67	0.122
		physical fitness	−12.031	4.458	−0.481	−2.698	0.010 *	−21.06	−3.00	

Final models of the backward linear regression analysing sub-domains-concept as dependent variable and the independent factors sex, age, BMI-SDS, adolescent status (<13 or ≥13 years old), migration background, media consumption in h/day, and physical fitness in watts/kg bodyweight in the different time periods (*n* = 205). * significant results (*p* < 0.05).

**Table 7 children-10-00127-t007:** Linear regression analysis (backward) of the global self-worth domain.

Domain	Periods of Analysis of Final Model		Unstandardized Coefficients		Standardized Coefficients	T	*p*-Value	95.0% Confidence Interval for B		Adjusted R^2^
			B	std. error	ß			lower limit	upper limit	
Global self-worth	2005–2006	media consumption	−6.380	2.498	−0.428	−2.554	0.016 *	−11.49	−1.27	0.155
	2007–2008	BMI-SDS	−9.828	6.516	−0.247	−1.508	0.140	−23.06	3.40	0.034
	2009–2011	adolescent status	7.532	4.779	0.290	1.576	0.127	−2.27	17.34	0.050
	2012–2014	sex	−8.326	4.857	−0.272	−1.714	0.100	−18.37	1.72	0.493
		age	−9.238	1.826	−1.482	−5.060	<0.001 *	−13.02	−5.46	
		adolescent status	33.217	7.893	1.075	4.209	<0.001 *	16.89	49.54	
		BMI-SDS	−23.491	5.799	−0.727	−4.051	<0.001 *	−35.50	−11.50	
		migration background	−21.038	6.180	−0.549	−3.404	0.002 *	−33.82	−8.25	
		media consumption	8.509	2.747	0.484	3.098	0.005 *	2.83	14.19	
	2015–2017	BMI-SDS	−17.232	5.891	−0.433	−2.925	0.006 *	−29.17	−5.30	0.166
	2018–2020	migration background	5.534	6.452	0.138	0.858	0.396	−7.53	18.60	−0.007

Final models of the backward linear regression analysing sub-domains of self-concept as dependent variable and the independent factors sex, age, BMI-SDS, adolescent status (<13 or ≥13 years old), migration background, media consumption in h/day, and physical fitness in watts/kg bodyweight in different time periods (*n* = 205). * significant results (*p* < 0.05).

**Table 8 children-10-00127-t008:** Linear regression analysis (backward) of the behavioural conduct domain.

Domain	Periods of Analysis of Final Model		Unstandardized Coefficients		Standardized Coefficients	T	*p*-Value	95.0% Confidence Interval for B		Adjusted R^2^
			B	std. error	ß			lower limit	upper limit	
Behavioural conduct	2005–2006	adolescent status	−10.512	4.556	−0.328	−2.307	0.029 *	−19.85	−1.18	0.413
		physical fitness	23.668	6.312	0.533	3.750	0.001 *	10.74	36.60	
	2007–2008	age	−1.160	1.437	−0.135	−0.808	0.425	−4.08	1.76	−0.010
	2009–2011	adolescent status	8.297	5.166	0.295	1.606	0.120	−2.30	18.90	0.053
	2012–2014	sex	−16.685	5.841	−0.478	−2.856	0.009 *	−28.77	−4.60	0.435
		age	−9.977	2.196	−1.405	−4.544	<0.001 *	−14.52	−5.44	
		adolescent status	31.483	9.493	0.894	3.317	0.003 *	11.85	51.12	
		BMI-SDS	−13.831	6.975	−0.376	−1.983	0.059	−28.26	0.60	
		migration background	−17.257	7.433	−0.396	−2.322	0.029 *	−32.63	−1.88	
		media consumption	10.595	3.304	0.529	3.207	0.004 *	3.76	17.43	
	2015–2017	migration background	−20.648	7.727	−0.402	−2.672	0.011 *	−36.31	−4.99	0.139
	2018–2020	migration background	−10.326	6.436	−0.252	−1.604	0.117	−23.35	2.70	0.039

Final models of the backward linear regression analysing sub-domains of self-concept as the dependent variable and the independent factors sex, age, BMI-SDS, adolescent status (<13 or ≥13 years old), migration background, media consumption in h/day, and physical fitness in watts/kg bodyweight in different time periods (*n* = 205). * significant results (*p* < 0.05).

## Data Availability

The data used and analysed during the current study involve sensitive patient information and indirect identifiers. As a result, the datasets are available from the corresponding author only on reasonable request.

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
