# Peer review of "Secular Trend of Self-Concept in the Context of Childhood Obesity—Data from the CHILT III Programme, Cologne"

_children, 2023, doi:10.3390/children10010127_

Round 1
Reviewer 1 Report (Previous Reviewer 1)
First of all I would like to congratulate the authors, as I think they have done a great job. The manuscript is of high quality. However, I have some comments that should be resolved:
There are major typos in the formatting (keywords, reference list, table footnotes...).
Line 77, BMI-SDS, first time in main text, express.
In the tables it is recommended to replace "*" and "**" with other nomenclature as it may confuse the reader, as these symbols are often used to define the level of significance.
In general the results are discussed very briefly, and should be contrasted with more studies on the subject, as it is a well-studied topic.
Line 341, describes applications and future directions, which should be expressed and specified in the discussion, thus providing a higher quality.
Author Response
First of all I would like to congratulate the authors, as I think they have done a great job. The manuscript is of high quality.
Thank you very much for this appreciative comment.
However, I have some comments that should be resolved:
There are major typos in the formatting (keywords, reference list, table footnotes...).
Thank you for reading carefully. Accordingly, we have completely revised the manuscript again.
Line 77, BMI-SDS, first time in main text, express.
Thank you very much. We have resolved the abbreviation.
In the tables it is recommended to replace "*" and "**" with other nomenclature as it may confuse the reader, as these symbols are often used to define the level of significance.
Thank you also for this comment. We have replaced the characters accordingly.
In general the results are discussed very briefly, and should be contrasted with more studies on the subject, as it is a well-studied topic.
Thank you for these valid comments. We have gone into more depth in the discussion and the conclusion.
Line 341, describes applications and future directions, which should be expressed and specified in the discussion, thus providing a higher quality.
Please see above
Reviewer 2 Report (Previous Reviewer 2)
This is very intersting and well performed study. The structure of the project ,statistical analysis, discussion and conclusions are on thevery good level. The abstract is to long and should be improved.
The authors should explain and writing more about definition of adolescent status which used in manuscript.It would be great to known if were differences in the results during such long periods between boys and girls.
I would like to recommned this manuscript for publication after suggested by me minor changes changes.
Author Response
This is very interesting and well performed study. The structure of the project, statistical analysis, discussion and conclusions are on the very good level.
Many thanks for this appreciative comment, too.
The abstract is too long and should be improved.
Thank you very much. We have significantly shortened and streamlined the abstract.
The authors should explain and writing more about definition of adolescent status which used in manuscript. It would be great to known if were differences in the results during such long periods between boys and girls.
Thank you. That is a very important aspect. We have better elaborated the definition in the method section. The point was also taken up again in the discussion. However, there are no clear differences in our age groups. In order to be able to make relevant statements on this very exciting and important point, it would be better to assess the Tanner stage and we would probably need larger groups.
I would like to recommend this manuscript for publication after suggested by me minor changes.
Many thanks.
This manuscript is a resubmission of an earlier submission. The following is a list of the peer review reports and author responses from that submission.
Round 1
Reviewer 1 Report
First of all, I would like to congratulate the authors for their work.
This is a very complete, concise and very specific paper. Overall, the linguistic expression is good, direct and fluent. The content is very rich and mostly scientifically supported. I also find that it has great practical application since, as the authors point out, it can help in the more specific orientation of therapeutic programmes in this population.
However, I have some minor comments:
- There is a lack of a sample calculation to know if the sample described is sufficient for the results to be consistent.
- Section 2.1. it is recommended to include specific inclusion and/or exclusion criteria.
- What instruments were used for weight and height measurements?
- Section 3.1. was this increase significant? Specify.
- Where reference is made to "(cf. Tab. 2)" or "(cf. Tab. 1)", it is recommended that these be changed to "(Tab. 2)", as the "cf" could confuse the reader.
- It is recommended to improve the formatting of the tables, e.g. start variables and domains with capital letters. "age" for "Age". As well as adjusting and centring the cells.
Reviewer 2 Report
This is an interesting manuscript. However, the topics it covers were previously described in the literature.
The research methodology is well planned, but its main disadvantage is the small number of people surveyed in each group. The analysis of such small groups, especially in the context of psychological research, allows not for drawing far-reaching conclusions.
Therefore, although the manuscript is well written and structured, the work is not suitable for publication.
Reviewer 3 Report
This is an assessment of self concept secularly (not longitudinally) at six times from 2005-6 to 2018-20 among children enrolled in obesity treatment programs. The authors make spurious claims (increase in overweight or obesity); they conduct a very large number of tests each with very small samples n≈40) yet retain p<0.05 as their significance criterion; there was huge sample attrition (50%) from originally enrolled sample.
The title of this manuscript is misleading. Self concept should replace psychological distress. The authors did not indicate what level of change in self concept would indicate distress. This is particularly a concern in light of there being no differences in any of the self concept variables over time.
To assess possible participation bias, please provide a table that displays the demographic and anthropometric characteristics of the integrated sample (n=242) versus the non-integrated sample (n=293) with the available data. Would a data imputation for missing data be appropriate?
Please provide a rationale for reporting sex differences by demographic and anthropometric characteristics in Table 1.
Can the authors provide more demographic characteristics, e.g. socioeconomic status, family characteristics, migration status in Table 1?
Please provide the question(s) used to assess media consumption and add the data to Table 1.
Accepted use suggests the term sex is preferred to gender, which assesses self identity.
How was pubertal status assessed?
Results: The authors report a difference in BMI from 29.5 at Time 1 to 30.5 at Time 6 as if it were significant, but their data show the complete lack of significance p=0.867. Not even close. If it is not significant, it is not significant. Attention should not be drawn to a non-significant difference.
In the Discussion, they say that BMIs tended to increase. This is false for their data. This is contradictory to secular data by time in other data sets, but the participants here were already overweight or obese, which may have influenced the lack of difference by time.
The authors report an extensive number of backward elimination regression tests predicting each of the six self concept variables for each of the time intervals. The sample sizes are way too small to have confidence in these results (e.g. n=34). There is little consistency in the patterns of significant effects over time. There appeared to be no time related trend in which variables were predictive. This analysis might be appropriate for the full sample (n=242) or even the large sample (n=535) using mixed models which can conduct analysis even with missing data, using time as a predictor.
The Discussion will change if the analyses change.
How representative of overweight and obese children in Cologne is this sample (n=535) of children obtaining care in this trial? Can the authors compare the demographics of their sample with population demographics usually obtained by government agencies? is there evidence of participation bias that would influence an interpretation of the results?